# Cyclin C: The Story of a Non-Cycling Cyclin

**DOI:** 10.3390/biology8010003

**Published:** 2019-01-04

**Authors:** Jan Ježek, Daniel G. J. Smethurst, David C. Stieg, Z. A. C. Kiss, Sara E. Hanley, Vidyaramanan Ganesan, Kai-Ti Chang, Katrina F. Cooper, Randy Strich

**Affiliations:** Department of Molecular Biology, School of Osteopathic Medicine, Rowan University, Stratford, NJ 08084, USA; smethurstd@rowan.edu (D.G.J.S.); stiegdc@rowan.edu (D.C.S.); kissz55@rowan.edu (Z.A.C.K.); hanleys2@rowan.edu (S.E.H.); vidu.biologist@gmail.com (V.G.); changk@rowan.edu (K.-T.C.); cooperka@rowan.edu (K.F.C.); strichra@rowan.edu (R.S.)

**Keywords:** cyclin family, transcriptional cyclins, Cdk8-dependent kinase module, Mediator, stress signaling, yeast, tumor suppressor, cancer

## Abstract

The class I cyclin family is a well-studied group of structurally conserved proteins that interact with their associated cyclin-dependent kinases (Cdks) to regulate different stages of cell cycle progression depending on their oscillating expression levels. However, the role of class II cyclins, which primarily act as transcription factors and whose expression remains constant throughout the cell cycle, is less well understood. As a classic example of a transcriptional cyclin, cyclin C forms a regulatory sub-complex with its partner kinase Cdk8 and two accessory subunits Med12 and Med13 called the Cdk8-dependent kinase module (CKM). The CKM reversibly associates with the multi-subunit transcriptional coactivator complex, the Mediator, to modulate RNA polymerase II-dependent transcription. Apart from its transcriptional regulatory function, recent research has revealed a novel signaling role for cyclin C at the mitochondria. Upon oxidative stress, cyclin C leaves the nucleus and directly activates the guanosine 5’-triphosphatase (GTPase) Drp1, or Dnm1 in yeast, to induce mitochondrial fragmentation. Importantly, cyclin C-induced mitochondrial fission was found to increase sensitivity of both mammalian and yeast cells to apoptosis. Here, we review and discuss the biology of cyclin C, focusing mainly on its transcriptional and non-transcriptional roles in tumor promotion or suppression.

## 1. Introduction

Cyclins, so named as this eukaryotic protein family displayed cycling expression patterns throughout the cell cycle [1], formed the basis of research resulting in a broad understanding of cell cycle control [2]. The groundbreaking initial work revealed an oscillating expression pattern set up by synthesis and subsequent degradation of cyclin proteins. Cyclins act in partnership with cyclin-dependent serine/threonine protein kinases (Cdks) whose activity also peaks during specific cell cycle stages as a result of the cyclin interaction. When active, cyclin-Cdk complexes drive cell cycle progression by phosphorylating proteins that play vital roles in multiple cellular processes including proliferation, differentiation, metabolism, and homeostasis [3]. We refer to the cell cycle regulators as class I cyclins (Table 1). Altogether, the class I comprises at least 20 Cdks and 30 cyclin proteins, which are ubiquitously expressed (Table 1) [4]. In addition, cyclins and their Cdk partners have many functions other than cell cycle control including proteolytic degradation, programmed cell death, DNA damage repair, metabolism, stem cell self-renewal, and spermatogenesis [5]. This group can be divided into non-cycling transcription factors (class II) and other (class III) cyclins (Table 1).

There are several common features of the class II transcriptional cyclin-Cdks. For example, these cyclin-Cdks associate with the RNA polymerase II (Pol II) holoenzyme to control gene-specific transcription [8]. In addition, the cyclin-Cdk complexes of cyclin H-Cdk7, cyclin C-Cdk8, and cyclin T-Cdk9 all phosphorylate the C-terminal domain (CTD) of Pol II to stimulate or inhibit its activity. One of these complexes, cyclin C-Cdk8, which is well conserved among eukaryotes, is the focus of this review [9]. The human and *Drosophila* cyclin C family members were discovered in independent yeast studies searching for activities that complement the cell cycle arrest phenotype associated with loss of all three G1 cyclins (*CLN1*–*3*) [10,11,12]. The first clue that cyclin C was different was its expression profile. Both at the mRNA [11] and protein [13] levels, cyclin C shows a shallow oscillatory pattern during cell cycle progression with a broad peak appearing in the G_1_ phase. Although cyclin C was the first “non-cycling” cyclin, later it was found that this expression profile was shared by other transcriptional cyclins such as cyclin H [14] and T [15].

In order to fulfill its duty as a transcription factor (referred to here as the day job function), cyclin C interacts with Cdk8 as part of a highly conserved multi-protein regulatory complex called Mediator that associates with RNA Pol II [16,17,18]. The complex between transcription factors, Mediator, and the transcription machinery is collectively termed as the RNA polymerase II holoenzyme [19]. Mediator also serves both as a signal-transducing and physical bridge between distant regulatory elements such as enhancers or super-enhancers and RNA Pol II. The complex between transcription factors, Mediator, and the transcription machinery is collectively termed as the RNA polymerase II holoenzyme [19]. In this large signal processing hub, cyclin C and Cdk8, together with accessory subunits Med12 and Med13, constitute an independent and reversibly dissociable regulatory sub-complex, referred to as the Cdk8-dependent kinase module (CKM) [20,21]. In yeast, the principal function of the CKM is to repress transcription by a variety of mechanisms including phosphorylating the CTD of Pol II [22,23]. On the contrary, the CKM also activates transcription in mammalian cells by phosphorylating cyclin H [24] and the transcriptional activator E2F [25]. Hence, the detailed underpinnings of the two-way mechanism of how cyclin C-Cdk8 integrates different transcriptional cues into changes in transcription via the Mediator remain a topic of intense investigation.

Mechanistic insight into RNA Pol II-mediated control is important as transcriptional dysregulation is one of the main drivers of cancer [18] and developmental diseases [26]. The variability in Mediator subunit composition due to expression differences or mutational status may be critical for the neoplastic growth initiation or developmental defects [27]. Indeed, Cdk8 overexpression has been associated with poor prognosis in cancer patients [27,28]. However, deletions of the genomic locus on chromosome 6 (6q21) harboring the gene encoding cyclin C (*CCNC*) have been identified in both solid tumors [29,30] and hematologic malignancies [31,32]. Therefore, despite the pro-oncogenic effects associated with Cdk8 overexpression, these observations allude to a tumor suppressive role of cyclin C. This inconsistency may be explained by cell-type specific differences in cyclin C-Cdk8 transcriptional control. An alternative possibility to explain this conundrum may stem from the non-canonical role that cyclin C plays as a key mediator of stress signaling [33].

In this scenario, the stress signaling role of cyclin C can be fulfilled within the nucleus by de-repression of stress response genes, referred to here as canonical or a day job function. More intriguingly, cyclin C functions outside of the nucleus, referred to here as non-canonical or a night job function. Upon oxidative stress, cyclin C translocates into the cytosol where it binds and activates the mechanoenzyme dynamin-related protein 1 (Drp1) to induce mitochondrial fission [34]. An abrupt shift from elongated to fragmented mitochondrial phenotype is believed to serve as a signaling platform for the activation of either pro-survival or pro-death stress response mechanisms such as mitophagy or intrinsic apoptosis, respectively, with mild stress activating the former and severe stress triggering the latter [33].

In the following paragraphs, we summarize the current knowledge on cyclin C biology including its dual role in eliciting stress signaling while highlighting potential implications relevant to cancer and cancer therapy.

## 2. Phylogeny of the Cyclins

### 2.1. Sequence Features within the Cyclin Family

Cyclins are part of a conserved group of proteins that, with few exceptions, contain two characteristic cyclin box fold domains (Figure 1). The cyclin fold which consists of approximately 100 residues forming a five-helix bundle [35]. Typically, the C-terminal cyclin box is required for correct protein folding, while the N-terminal domain binds the Cdk. The cyclin box fold is also present in other proteins, which do not appear to activate Cdks including TFIIB and retinoblastoma (Rb) [6]. Cyclins can be phylogenetically divided into three classes, with those that function in transcriptional regulation falling into a class that includes cyclin C, as well as cyclins H, K, L, M, and T (Table 1) [36]. Further small regions of homology beyond the cyclin box are also observed between some cyclin proteins, representing additional functional motifs [37].

### 2.2. Phylogenetic Analysis of Cyclin C Sequences

A phylogenetic tree was constructed to understand the divergence of cyclin C in different species as well as differences amongst members of the cyclin family (Figure 2). This tree is mapped using Grishin distance which estimates the distance of homologous proteins based on the number of amino acid substitutions per site, therefore the further the distance between the two proteins the more substitutions have occurred [39]. Surprisingly, the transcriptional cyclins K and T segregated early and are, therefore, more distant to cyclin C than the cycling cyclins A, D, and E suggesting that early predecessors of these two transcriptional cyclins represented the primordial ancestry of the whole cyclin C family. This result is different than phylogenetic studies that used only conserved regions such as the N- and C- terminal domains to align the cyclin family. For example, using this selective study, Cao et al. found that cyclin C was most related to the other transcriptional cyclins K, H, and T and this group developed independently from cell cycle cyclins [40]. Our analysis reveals an unprecedented relatedness between these two cyclin classes and opens the question of potential redundancy among cyclins A, D, E, K, and T in their capacity to activate Cdks. Previous studies have noted that deleting all three D-type cyclins or the two cyclin E isoforms that reported control G1 progression or the G1-S phase transition are largely dispensable for mouse development [41], leading to the notion that other cyclin family members can provide cell cycle regulatory functions. Our data provide a basis for speculation that cyclin K and T predecessors were early drivers of cell cycle progression but may have been supplanted in this role by evolving cyclins D and E. Under exactly what conditions, if any, that transcriptional cyclins can provide cell cycle-specific functions needs to be tested experimentally.

Cyclin K partners with Cdk9 to respond to replicative stress, whereas the cyclin T-Cdk9 kinase complex regulates RNA Pol II elongation [42]. Cyclin H is another transcriptional cyclin that is more closely related to cyclin C and associates with TFIIH as does cyclin C [43]. The distance between transcriptional cyclins could be explained by the different roles these cyclins play in transcription and stress response.

### 2.3. Anchoring of Cyclin C in the Nucleus is Remarkably Conserved

Cyclin C is the most conserved cyclin from yeast to man, sharing 28% sequence identity (Figure 3). Although Cdk8 is also well conserved from yeast to mammals, the other components of the CKM, Med12 and Med13 display a significant degree of structural, but not sequence, conservation. In addition to the cyclin box domains, the cyclin C family shares a unique and evolutionary conserved N-terminal α-helix. In cyclin C, this N-terminal α-helix is much shorter and mobile compared to other cyclins. This unique cyclin C α-helix can therefore adopt many different spatial positions, whereas other cyclins contain very rigid and integral α-helices that restrict their flexibility. This helix contains the holoenzyme association domain (HAD) which is required for Med13 binding and nuclear localization. HAD consists of a basic region containing a KERQK sequence located between the N terminus and the first α-helix of the first cyclin box domain of cyclin C (Figure 4) [44]. In addition, nuclear to cytoplasmic translocation of the yeast cyclin C is inhibited by mutations in Ala110 and Glu170 that form an interaction face when folded into its proper configuration [45,46], (see Figure 3). Glu170 is required for transcriptional function while Ala110 is not indicating that nuclear release is independent of its transcriptional function [47]. Whether these amino acids are required for nuclear release of the human cyclin C is yet to be determined.

The cyclin C-Cdk8 pair is unique in that Cdk8 contains a specific recognition α-helix for cyclin C that limits promiscuity with other cyclins. This unique region of Cdk8 is located within an N-terminal α-helix, αB, which together with another α-helix, αC, forms additional interactions with cyclin C outside of the common binding surface (Figure 4). The binding of cyclin C and Cdk8 is much tighter compared to other transcriptional Cdk-cyclin pairs such as Cdk9-cyclinT [48]. This exceptionally high affinity and specificity may be required for the recruitment of other CKM components such as Med12 and Med13. Lastly, cyclin C contains a unique, highly conserved groove which is located between the two cyclin repeats that may serve to interact with other CKM members [49].

## 3. The Transcriptional Function of Cyclin C-Dependent Kinases

Cyclin C and Cdk8 are both members of the CKM, along with Med12 and Med13. As part of the Mediator complex, the CKM is capable of modulating RNA Pol II-dependent transcription in both a positive and negative manner. Excluding the CKM, the Mediator core structure can be divided into head, middle, and tail domains [17,27,50]. In contrast to the CKM, however, these structural elements are not dissociable and form a single physical unit. Gene-specific transcription factors from multiple signal transduction pathways such as amyloid precursor-protein (APP) [51], Notch [31,52], Sonic hedgehog (SHH) [53,54], and wingless/integrated (Wnt)/β-catenin [55,56,57,58] pathways associate with the Mediator through the CKM or the tail domain. The signal is then propagated by the middle domain to the head domain, which connects the assembled Mediator complex with RNA Pol II to impact transcription. In *C. elegans*, Mediator is also the target of the epidermal growth factor receptor (EGFR)/Ras/mitogen-activated protein kinase (MAPK) signaling pathway [59]. In addition, Cdk19, a closely related paralog of Cdk8 sharing a 77% primary sequence identity [60], may theoretically substitute for the transcriptional regulatory role of Cdk8 in the CKM. Similarly, Med12L and Med13L are close paralogs of Med12 and Med13, respectively, and hence may represent an interchangeable alternative in terms of structure and function [18].

The yeast cyclin C-Cdk8 kinase complex plays a key role in the transcriptional repression of a plethora of stress response genes [8,44,61,62] such as catalase [21] or the heat shock protein 70 (Hsp70) family member Ssa1 [47]. Upon exposure to various forms of cell stress including oxidative or osmotic stress, ethanol treatment, heat shock, and nutrient deprivation or when grown on non-fermentable carbon, the yeast cyclin C is rapidly degraded in a controlled manner to relieve the transcriptional repression and hence to initiate a robust stress response [63]. The destruction of cyclin C occurs following its nuclear release into the cytoplasm, where it is marked by ubiquitination for degradation mediated by the 26S proteasome [45,46]. In summary, these results highlight the important role that cyclin C-Cdk8 kinase complex plays in the stress response by repressing genes whose transcription is required for survival following cellular damage. It remains to be determined whether the same strategy is in play in mammals.

## 4. The Mitochondrial Stress Signaling Pathway

### 4.1. Regulation of Mitochondrial Dynamics by Cyclin C

Mitochondria are a platform that orchestrates cellular responses to stress signals. Without any stress, mitochondria are predominantly in a connected and reticular morphology. The fused state allows for maximum ATP production and the repair of mitochondrial DNA or membrane damage. Physiologically, mitochondrial fission is activated during each cycle of mitosis to promote segregation of mitochondria between two daughter cells or as an anti-stress mechanism crucial for the removal of damaged mitochondria via mitophagy. Accordingly, mitochondria become extensively fragmented upon treatment with cytotoxic agents [64,65]. Extensive fragmentation of the mitochondrial network is an important first step in oxidative stress-mediated responses that lead to the release of pro-apoptotic factors from the intermembrane space in a process known as intrinsic apoptosis. Conserved dynamin-like GTPases, Drp1 in human and Dnm1 in yeast, play a key role in inducing mitochondrial fragmentation. Upon stress stimuli, Drp1/Dnm1 is recruited to outer mitochondrial membrane, where it oligomerizes into filament-like structures that encircle and cleave mitochondrial tubules in a GTP-dependent fashion.

The stress signaling function of cyclin C involves not only transcriptional but also non-transcriptional mechanisms as cyclin C was recently found to deliver the oxidative stress signal to mitochondria. In this pathway, cyclin C induces extensive mitochondrial fragmentation in both mammalian [34] and yeast [46,66] cells (Figure 5). In mammalian cells, oxidative stress induces a small fraction of cyclin C to be released from the nuclear pool into the cytosol as a second messenger signal. In the cytosol, cyclin C interacts with, and activates, the mitochondrial GTPase Drp1 to induce mitochondrial fission [34]. These data indicate that cyclin C induces mitochondrial fragmentation through the fission machinery. A similar mechanism was observed between yeast cyclin C and Dnm1 [66]. Although cyclin C mitochondrial relocalization and mitochondrial fragmentation was observed in HeLa cells, the human osteosarcoma U2O2 cells displayed nuclear cyclin C release and mitochondrial localization in the absence of oxidative stress but did not induce fragmentation [34]. Therefore, cyclin C’s control over mitochondrial dynamics may differ depending on the cancer type examined.

Analogously, the yeast ortholog of Drp1, Dnm1, mediates oxidative stress-induced mitochondrial fission in a cyclin C-dependent manner [66]. The CKM subunit Med13 was identified as the nuclear anchor of cyclin C since its deletion in yeast coincided with nuclear cyclin C release even in the absence of oxidative stress [67]. Subsequent studies in yeast have revealed that cyclin C nuclear release requires cell wall integrity sensors [68], the Slt2 MAP kinase signaling pathway and the AMP-activated protein kinase (AMPK) Snf1 [69,70,71]. Following oxidative stress, these pathways target Med13 for proteasomal-dependent degradation through *the S-phase kinase-associated protein (Skp)-*Cullin-F-box (SCF)-mediated ubiquitination of Med13 [72]. These experiments are consistent with a study performed in cancer cell lines, which shows that the substrate recognition component of SCF, F-box and WD repeat domain-containing 7 (Fbw7), elicits its tumor suppressive function by targeting Med13 and Med13L for degradation [73]. Given that Med13 serves as an efficient anchor for the retention of cyclin C within the nucleus, proteasomal destruction of Med13 may constitute an important step in the release of cyclin C from the nucleus. Nevertheless, whether a similar mechanism contributes to the release of cyclin C upon oxidative stress in higher eukaryotes remains to be investigated.

### 4.2. Regulation of Apoptosis by Cyclin C

Extensive fragmentation of the mitochondrial network is considered an important first step in oxidative stress-mediated responses that lead to the release of pro-apoptotic factors from the intermembrane space during the process of intrinsic apoptosis [74,75]. Since evasion of apoptosis is one of the hallmarks of cancer, the direct role of cyclin C at the mitochondria could straightforwardly explain its observed tumor suppressive activity. Indeed, wild-type, but not CCNC knock-out mouse embryonic fibroblast (MEF) cells, were susceptible to cisplatin-induced apoptosis [34]. The effect positively correlated with proportional increase in the fragmented mitochondrial phenotype observed in cisplatin-treated wild-type but not CCNC knock-out MEF cell lines, wherein it could be rescued by ectopic expression of cyclin C. However, this activity is specific for mitochondrial-dependent cell death as cyclin C was not required for death receptor-mediated, or extrinsic, apoptosis. Moreover, wild-type, but not CCNC knock-out H_2_O_2_-treated MEF cell cultures, displayed mitochondrial dysfunction assessed by the dissipation of mitochondrial membrane potential, hinting at the possibility that mitochondrial reactive oxygen species (ROS) may constitute a common denominator in cyclin C-dependent cell killing. Analogically, cyclin C is required for the sensitivity of yeast cell cultures to oxidative stress, measured as plating efficiency and terminal deoxynucleotidyl transferase dUTP nick end labeling (TUNEL) positivity [67]. Although these findings may have important implications for cancer research as well as clinical applications, the exact molecular mechanism of cyclin C-dependent activation of the apoptotic machinery deserves future investigation.

## 5. The Structure of Cyclin C and Its Links to the Mediator

### 5.1. Structural Insights into the Day Job Function of Cyclin C

#### 5.1.1. Cyclin C Crystal Structures

Similar to practically all cyclins, cyclin C possesses two 5 α-helix cyclin boxes (Figure 4). Unlike most other cyclins that have additional amino and carboxyl α-helical regions (H_N_ and H_C_), C-type cyclins lack the C-terminal domain [49]. Although the human and yeast cyclin C share many structural elements and key residues, differences are noted. For example, the human H_N_ α-helix is shorter and makes contact with Cdk8. In the yeast cyclin C-Cdk8 crystal structure, the longer H_N_ α-helix does not interact with Cdk8, although it is thought to be mobile. The human cyclin C further contacts Cdk8 via the H_C_ α-helix, which is absent in the *S. pombe* homolog [48]. While an N-terminal α-helix is seen in the other cyclins, in cyclin C it is not packed into the cyclin box α-helices and is mobile. Cyclin C also differs at the C terminus, as it has no additional α-helices following the second cyclin box repeat [66].

A highly conserved groove has been described which is specific to cyclin C family members [49]. Five residues at the surface of this structure are conserved in all C-type cyclins (Ile42, Arg58, Trp177, Asp182, and Tyr184) (Figure 3 and Figure 4). The groove region is negatively charged and distant from the Cdk8 interface suggesting that this region is involved in Med12 or Med13 binding. Comparison of crystal structures of *S. pombe* cyclin C with human cyclin H (PDB: 1JKW) [76] or cyclin A (PDB: 1FIN) [77] shows structural similarity based on the two cyclin box repeats.

#### 5.1.2. The Cyclin C and Cdk8 Interface

General principles of Cdk-cyclin binding have been established through studies of other members of this family, in particular Cdk2-cyclin A [78]. Cdks typically contain a regulatory, reversibly phosphorylated region called the T-loop that can block the active site, and an α-helix containing a PSTAIRE amino acid motif. In Cdk8, the PSTAIRE-like sequence (S(M/Q)SACRE) is implicated in cyclin binding. In the inactive form of Cdk2, the T loop and an adjacent small α-helix (L12) blocks the active site cleft. This conformation also excludes the α-helix containing the PSTAIRE motif which contains Glu51, a residue important for coordinating ATP. Cyclin A binding causes the T-loop to move out of the active site. Additionally, the L12 α-helix switches to a β-strand structure, and Glu51 moves away allowing reorientation of the ATP molecule. The T-loop becomes phosphorylated and interacts with cyclin A, and in this conformation, a substrate can interact with residues in the T-loop to become phosphorylated [78,79]. However, phosphorylation of the T-loop is not required for human Cdk8 since the usual phosphorylation target threonine residue is absent, suggesting a different activation mechanism. As Cdk8 possesses an aspartate (Asp176 in *S. pombe*) in the T-loop region, is has been speculated that this substitution allows Cdk to bypass the T-loop phosphorylation event [49]. Hence, changes to the amino acid sequence in the T-loop segment appear to have circumvented the necessity for activation of Cdk8 by phosphorylation [8].

#### 5.1.3. The Cdk8-Dependent Kinase Module

The CKM subunits in yeast were identified as negative regulators of transcription [47,80,81,82], and the C-terminal domain of RNA Pol II was found to be a phosphorylation target of Cdk8 [83]. Accordingly, phosphorylation of Pol II CDT by Cdk8 may be required for the transcriptional repression elicited by the CKM [80,84]. Med13 and Med12 are required, and the CKM interacts with the tail domain of Mediator via Med13. This interaction occurs prior to RNA Pol II recruitment [85]. Further attempts to elucidate the mechanism of RNA Pol II-dependent transcription in yeast came from an elegant study by Tsai et al. [86]. The authors deployed biochemical and electron microscopy tools to delineate the role of the middle domain of the Mediator in mediating the negative regulatory effects of the CKM on RNA Pol II. These results confirm that deciphering the structure-function relationship in RNA Pol II-mediated transcription is still an area of active research and future studies will be required for gaining more comprehensive understanding of the versatile function in regulating this process.

#### 5.1.4. Structure of the Mediator Complex

The Mediator complex was discovered in yeast as a multi-subunit complex which acts as an adaptor to bring together RNA Pol II with both basal and regulatory transcription factors [87] and is essential in all eukaryotes. Mammalian and yeast Mediator is composed of up to 30 and 25 subunits, respectively, comprising the head, middle, and tail modules, plus the CKM that is present in sub-stoichiometric concentrations and can dissociate [18,88]. RNA Pol II is responsible for transcription of all protein-coding RNAs, as well as most non-coding genes [20]. The Mediator complex does not directly bind DNA and interacts directly with RNA Pol II only transiently during preinitiation, dissociating before elongation [89]. Physical interactions also occur between the Mediator complex and both general and specific transcription factors [20].

Transcription factors bind at specific DNA sequences on promoter and enhancer regions to regulate gene transcription. These factors generally function to activate other transcriptional regulators, including the Mediator complex. The Mediator complex can interact with multiple transcription factors simultaneously, via interactions with its multiple subunits. Furthermore, Mediator regulates the activity of RNA Pol II via direct interactions, and in this way Mediator is able to relay and integrate the binding of transcription factors at a regulatory sequence to the transcription machinery. Mediator is important for transcriptional initiation and elongation, as well as regulating the topological organization of genomic DNA. While there is a typical subunit complement for a Mediator complex, different transcription factors interact with different subunits, and for certain roles the complex can apparently function with fewer component proteins. In some differentiated cells, core subunits may be dispensable, perhaps reflecting the expression of a restricted gene subset [79].

### 5.2. Structural Insights into the Night Job Function of Cyclin C

Translocation of cyclin C from the nucleus to the cytosol induces mitochondrial fission through Drp1 and the mitochondrial fission machinery [34]. What is the mechanistic basis of activation of Drp1 by cyclin C? Drp1 consists of the N-terminal GTPase domain, middle stalk domain that facilitates Drp1 oligomerization, and the C-terminal GTPase effector domain (GED). An unstructured region called the variable domain (VD) within the GED is critical for membrane lipid interactions of Drp1 [90]. In contrast to the small GTPases involved in cell signaling, Drp1 has very low basal GTPase activity and low affinity for GTP both of which are stimulated under conditions that facilitate higher-order assembly of Drp1 such as by cardiolipin. We have found that cyclin C binds the N-terminal GTPase domain of Drp1 and enhances the affinity of Drp1 to GTP to stimulate its GTPase activity [91]. Cyclin C binding to Drp1 in the presence of non-hydrolyzable GTP analog β,γ-methyleneguanosine 5′-triphosphate (GMP-PCP) induces formation of longer filaments by disassembling non-productive Drp1 rings and small filaments. These longer Drp1 filaments display enhanced activity. Consistent with this, mutants of Drp1 that cannot form higher-order oligomers do not show increased activity in the presence of cyclin C. The interaction between cyclin C and Drp1 is mediated through the C-terminal cyclin box domain of cyclin C independently of the N-terminal cyclin box, demonstrating that the night job (cytosolic functions of cyclin C) and day job (transcriptional role in the nucleus) are structurally bifurcated. Indeed, heterologous expression of the C-terminal cyclin box and not the N-terminal cyclin box is sufficient to restore the ability of *CCNC* knock-out MEF cells to undergo mitochondrial fission upon oxidative stress. This is the first demonstration of a molecular function for the C-terminal cyclin box for any cyclin, though this region is found in almost all cyclin family proteins and could be a frontrunner for exploration of the molecular basis of the non-canonical functions the cyclin family.

Scientists have debated the stoichiometry of Drp1 oligomers that promote interaction with other members of the mitochondrial fission machinery. Some researchers argue that Drp1 dimers interact with mitochondrial fission factor (Mff) and form oligomers on the mitochondrial surface [92,93], while others have favored a model in which Drp1 forms oligomers in the cytosol and these structures interact with Mff to induce mitochondrial fission [93,94]. Our results support a model of Drp1 activation, in which cyclin C induces disassembly of non-productive oligomers of Drp1 into dimers and facilitates their reassembly into fission-competent filaments [91].

## 6. Mechanisms of Tumor Suppression by Cyclin C

The duality of cyclin C function is perhaps best illustrated by its suppressing and enhancing role in cancer development. Early indications that cyclin C may function as a tumor suppressor came from loss of heterozygosity studies that found that the *CCNC* locus (6q21) is associated with loss or translocation in approximately 10% of cases of childhood acute lymphoblastic leukemia (T-ALL), with concomitant loss of one *CCNC* allele in over 90% of these cases [95]. Deletion in the proximal arm of 6q was also observed in one third of prostate cancers, with over 75% of these cases exhibiting loss of markers within 6q16.3–q23.2 [96]. Chromosomal loss in 6q is also seen in osteosarcoma, with loss of markers at 6q21 detected in 61% of observed cases [30].

Proper cyclin C signaling is integral to biological development, as full knockouts were embryonic lethal in mice, and resulted in pronounced histopathologic abnormalities in the murine placenta. Yet, conditional cyclin C knockout mice remained viable, albeit exhibiting hyperplasia in induced thymocytes and B cells. Furthermore, cyclin C knockdown in MEF cells did not significantly alter cell growth, transcription, or fractional distribution of cells within phases of the cell cycle; however, it did impair cell cycle reentry [31]. These results underscore the complexity of cyclin C signaling, as perturbing its expression results in defects in both embryonic development and tumor suppression.

Quiescent stem cell populations exist in distinct topologic niches, defined by features of the extracellular environment (e.g., neighboring cells, secreted factors, etc.); in the context of cancer stem cells (CSCs), this is termed the tumor microenvironment. CSCs are defined by their inherent capacity for self-renewal, tumor initiation, and repopulation. Additionally, CSCs are known to exhibit a high degree of phenotypic plasticity, demonstrating the ability to revert to a stem-like state, and become refractory to therapeutic intervention. Enhanced CSC survival may promote recurrence in many cancers [97]. Indeed, implantation of isolated human breast cancer CSCs resulted in de novo tumor development in mice, highlighting the mechanistic rationale apropos the complete eradication of CSC populations as a clinical treatment outcome [98].

Notch signaling is critical to development, promoting the differentiation and self-renewal of stem cells. Typical Notch signaling is initiated when a transmembrane ligand on a contiguous cell is recognized by a Notch receptor on the effector cell. Subsequent proteolytic cleavage of the Notch receptor releases the Notch intracellular domain (NICD), which migrates to the nucleus and triggers transcription of genes involved in differentiation such as Hairy/Enhancer of split (HES) gene family. Over-activation of the Notch pathway is a driving force in many cancers [99]. Cyclin C-Cdk8 inhibits Notch1 signaling by phosphorylating and priming NICD for proteasomal degradation via SCF-Fbw7 ubiquitin-ligase [52] (Figure 6). Indeed, full or partial ablation of cyclin C increased ICN1 levels in hematopoietic progenitor cells (HPCs), thymocytes, and MOLT-16 cells. The same study investigated DNA sequences at the *NOTCH1* locus from 73 patients with T-ALL and found point mutations in five cases which demonstrated decreased phosphorylation and turnover. Furthermore, mice injected with HPCs harboring either *NOTCH1* mutations or *CCNC* deletion exhibited an accelerated T-ALL phenotype [31].

## 7. Cyclin C-Cdk8 as an Oncogene

With regards to an oncogenic role for Cdk8, the CKM positivity regulates several oncogenic signaling pathways. In 2008, work from Firestein et al. indicated the role of Cdk8 as an oncogene, which is upregulated in the vast majority of colorectal cancers [55]. This group demonstrated that the β-catenin pathway is upregulated by Cdk8. This finding implicated the CKM as an important positive upstream regulator of other oncogenes. Follow-up studies in patients with colorectal cancer demonstrated that upregulation of Cdk8 led to decreased survival [100]. More recently, angiogenesis in a pancreatic cancer model was enhanced by Cdk8 activation of the β-catenin-Krüppel-like factor 2 (KLF2) signaling axis [101]. In addition, Cdk8 has been shown to activate the oncogene YAP through direct phosphorylation, leading to colon cancer tumorigenesis [102]. Additionally, an inverse relationship was observed between Cdk8 activity and histone variant macroH2A (mH2A) levels. mH2A silences transcription and suppresses melanocyte malignancy. The data indicate that Cdk8 upregulation, combined with mH2A downregulation, accelerates melanomagenesis [103]. Cdk8 overexpression has been associated with stimulating the Myc oncogene and its downstream targets in colon cancer [104]. Cdk8 overexpression in breast cancer tumors also exhibits a strong positive correlation with Myc expression [105]. Importantly, this study also presented positive correlations between Cdk8 and both cyclin C and Med13 in breast cancers. Together, the current data indicate an important role for the CKM complex in the regulation of proliferation and oncogenic signaling in a variety of cancers.

In addition to playing role in cancer cells, Cdk8 demonstrates a positive regulatory role in normal cell types during differing conditions. Cdk8 is a positive regulator of several oncogenic transcription factors within the serum response network, including members of the early growth response (EGR) and activator protein 1 (AP-1) families [106]. In response to hypoxia, the CKM induces many HIF1α target genes [107]. The Espinosa’s group also found that HIF1α induces binding of Mediator-CKM to the super elongation complex to alleviate RNA Pol II pausing. This implicates a positive correlation between two oncoproteins, Cdk8 and HIF1α, in response to hypoxia. The importance of the CKM continues to be elucidated in a variety of conditions. The role of the CKM, specifically Cdk8 upregulation, in the variety of cancers and conditions presented by studies thus far implies a positive regulatory capacity driven toward proliferation. Since many tumor cells are proliferating in hypoxic conditions [108], it could be suggested that the upregulation of Cdk8 in cancer cells is preceded by the activation of HIF1α target genes by the CKM. Therefore, association between Mediator-CKM and HIF1α during hypoxia may lead to the induction of oncogenes and contribute to tumorigenesis. The potential for treatment of cancers directed toward the CKM and Cdk8 kinase activity is currently being investigated. Further investigation of the oncogenic capacity of Cdk8 will allow a better understanding of the function of the CKM during cancer development.

## 8. Pharmacological Targeting of Cyclin C-Cdk8

A positive correlation between Cdk8 upregulation and β-catenin hyperactivity initiated therapeutic approaches targeting Cdk8. However, due to their ubiquitous presence and rather shallow active sites, transcription factors are generally considered to be intractable drug targets for cancer therapy [109,110]. While numerous chemotherapeutic agents have been developed to inhibit the catalytic activity of Cdk8 (Table 2), attempts to directly target cyclin C have been limited. In this section, we outline some of the concepts in targeting Cdk8 for cancer therapy and provide an example of repurposing an already established drug to target the non-transcriptional mode of action of cyclin C.

Pharmacological screening for potent Cdk8 inhibitors is of utmost importance since, in line with its oncogenic role, Cdk8 has been identified as the down-stream effector of the Wnt/β-catenin pathway in colorectal adenocarcinoma [55,57,126,127]. The main challenge in targeting Cdks, in general, is to achieve selectivity between different Cdk family members [128]. In this respect, previous research focusing on ATP-competitive (type I) inhibitors failed due to a conserved topological landscape of the ATP-binding site among kinases. A more promising approach to develop Cdk-specific inhibitors avoids this issue by pharmacologically targeting an allosteric binding site with so-called type II inhibitors. Structurally, this site is represented by a hydrophobic groove residing near the ATP binding pocket. Allosteric control is mediated by conformational changes that occur within the T-loop region containing a highly conserved motif Asp173-Phe174-Gly175 (DFG). In this segment, the phenyl group of Phe174 plays a role of a flexible switch. In an active conformation (DFG-in), the phenyl ring of Phe174 occupies the hydrophobic groove so that Asp173 can facilitate ATP binding via two magnesium ions [129]. In the inactive conformation, or upon inhibitor binding into the hydrophobic groove (DFG-out), the phenyl ring of Phe174 becomes displaced several Angstroms into the ATP binding pocket and therefore sterically prevents nucleotide binding rendering the enzyme inactive. Unlike in other Cdks, the T-loop motif of Cdk8 is comprised of Asp173-Met174-Gly175 (DMG) [48]. For this reason, the hydrophobic groove of Cdk8 may be readily accessible to deep pocket (DMG-out) inhibitors with increased selectivity for Cdk8 among other kinases [111]. Indeed, the crystal structure of cyclin C-Cdk8 in the presence of sorafenib, which was also shown to interact with cyclin C-Cdk8 in a reporter displacement assay, revealed that this small molecule selectively binds to the allosteric pocket of Cdk8 in a DMG-out mode (Table 2) [111]. A follow-up structure-kinetic relationship study confirmed the association between Cdk8 and sorafenib and another known deep pocket-binding inhibitor BIRB796 [111]. Importantly, Senexin A [111] and B [124], discovered by the Roninson’s group, represent the new generation of selective Cdk8 inhibitors. Senexin A, discovered in a high-throughput screen for inhibitors of downstream targets of p21-dependent transcription in HT1080 fibrosarcoma cells [122], showed promising cytostatic activity in various cancer cell lines (Table 2). Senexin B, a more potent derivative of Senexin A, was found to synergize with fulvestrant to reduce the growth of breast xenograft tumors in mice [124], inhibit expansion of colon cancer metastases in the liver [125], and decrease the viability of leukemic cell lines [113]. Several other Cdk8 inhibitors were proven effective against acute or chronic forms of leukemia, colon and colorectal cancer (Table 2). The greatest remaining challenge, however, is to develop inhibitors that would discriminate between Cdk8 and Cdk19 isoforms whose kinase domains sequences are 97% identical [60].

A rare example of a drug targeting cyclin C independently of Cdk8 is the Hsp70 inhibitor pifithrin-μ [130], which was originally discovered as a specific inhibitor of mitochondrial p53 activity [131]. In analogy to p53, we have recently shown that pifithrin-μ blocks the transcription-independent activity as it diminished H_2_O_2_-induced co-localization of cyclin C with mitochondria in HeLa cells [34]. Consistent with the stress signaling role of cyclin C in activating mitochondrial fission and apoptosis, pifithrin-μ could be utilized as a cytoprotective agent in the treatment or prevention of degenerative diseases that are associated with altered mitochondrial dynamics [33]. A potential strategy for “activating” the mitochondrial role of cyclin C was provided by yeast studies. Deletion of the *MED13* anchor gene allowed constitutive cyclin C cytoplasmic localization [67]. Not only did this induce mitochondrial fragmentation, this also rendered the cell hyper-sensitive to oxidative stress. Therefore, a pharmacological method to disrupt cyclin C-Med13 interaction may allow sensitization of cancer cells to chemotherapy.

## 9. Conclusions

Cyclin C is rather an exceptional cyclin since its expression level does not significantly oscillate when compared to other cyclins. Compartmentalization of cyclin C between nucleus and the cytosol represents an important regulatory mechanism for executing an efficient response of cells to stress. This is achieved both at the transcriptional and non-transcriptional level by stress response gene de-repression and stimulation of mitochondrial fission and apoptosis, respectively. Owing to the essential role that cyclin C plays in stress signaling, future investigations into the direct function of cyclin C at the mitochondria may translate into new discoveries that would prove critical in combating both cancer and degenerative diseases.

## Figures and Tables

**Figure 1 biology-08-00003-f001:**
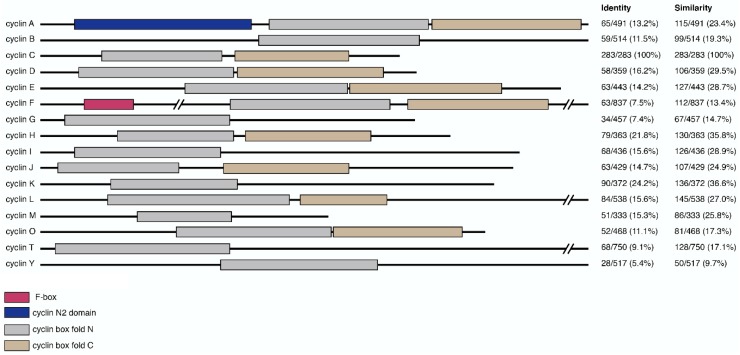
Domain composition of cyclin primary sequences and their relative similarity to cyclin C. Primary sequences of representative cyclins were annotated for domain-specific features including f-box (pink), cyclin N2 domain (blue), cyclin box fold N (gray) and C (brown). Relative sequence homology between cyclin C and other cyclins is shown on the right. Values, representing the number of identical or similar amino acid residues divided by the total protein length, were generated using EMBOSS Needle and expressed as percentages [38].

**Figure 2 biology-08-00003-f002:**
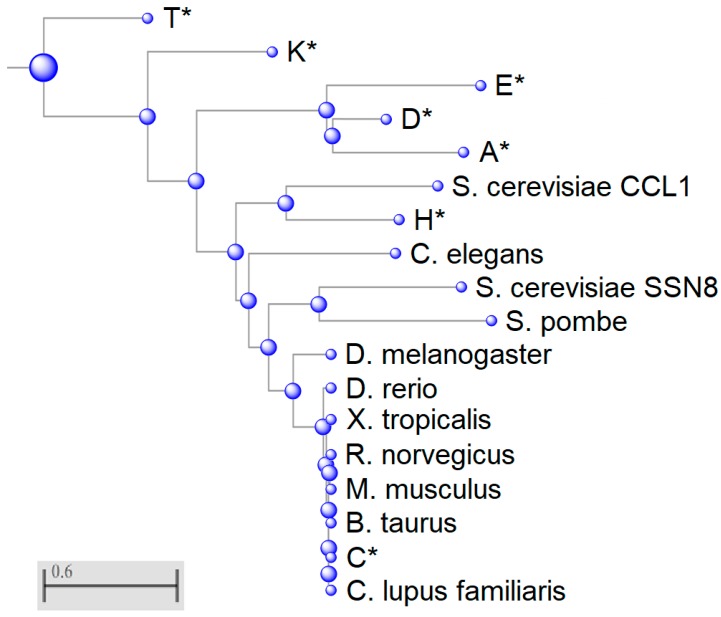
Phylogenetic analysis of eukaryotic cyclin sequences. Phylogenetic tree was generated using Basic Local Alignment Search Tool (BLAST). Members of the human cyclin family are indicated with an asterisk. All other sequences are cyclin C orthologs. *CCL1* is a yeast cyclin H paralog. *S. cerevisiae* Ume3/Srb11/Ssn8 and *S. pombe* cyclin C homologs are the most highly conserved cyclins within this multiple sequence alignment. Scale bar indicates number of amino acid substitutions per site.

**Figure 3 biology-08-00003-f003:**
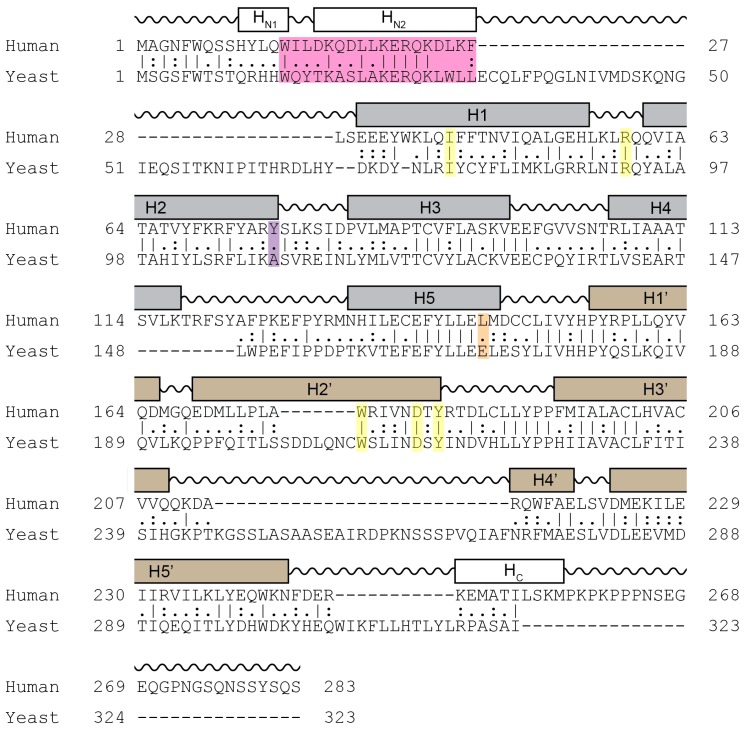
Sequence homology comparison between human and yeast cyclin C. Shown is the alignment of the primary sequences of human and yeast cyclin C. Grey and brown boxes denote α-helical segments of the N- and C-terminal cyclin box domains, respectively. Shown is the HAD domain (pink), the two amino acid residues Tyr76 (Ala110 in yeast) (magenta) and Leu145 (Glu170 in yeast) (orange) required for Cdk8 binding, and five highly conserved residues at the surface of a cyclin C-specific groove between the two cyclin box domains (human: Ile42, Arg58, Trp177, Asp182, and Tyr184) (yellow) (see Section 5.1.1) [47].

**Figure 4 biology-08-00003-f004:**
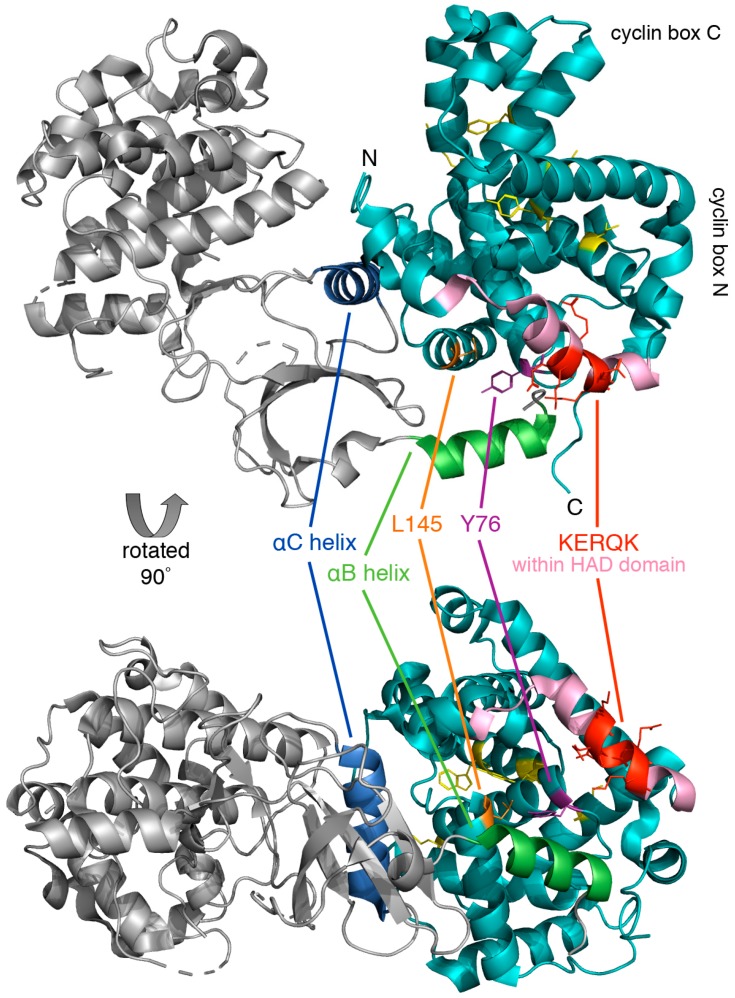
Crystal structure of human cyclin C-Cdk8 heterodimer. The ribbon model of cyclin C (cyan) and Cdk8 (grey) was generated using PyMol, PDB code 3RGF [48]. Highlighted is the conserved N-terminal KERQK sequence (red) within the HAD domain (pink) and two amino acid residues of cyclin C critical for establishing interaction with Cdk8, Tyr76 (Ala110 in yeast) (magenta) and Leu145 (Glu170 in yeast) (orange) [47]. The αB helix (green) and αC helix (blue) of Cdk8, which are responsible for making contact with the N-terminal cyclin box domain, and five highly conserved residues at the surface of a cyclin C-specific groove between the two cyclin folds (Ile42, Arg58, Trp177, Asp182, and Tyr184) (see Section 5.1.1) (yellow) are also indicated.

**Figure 5 biology-08-00003-f005:**
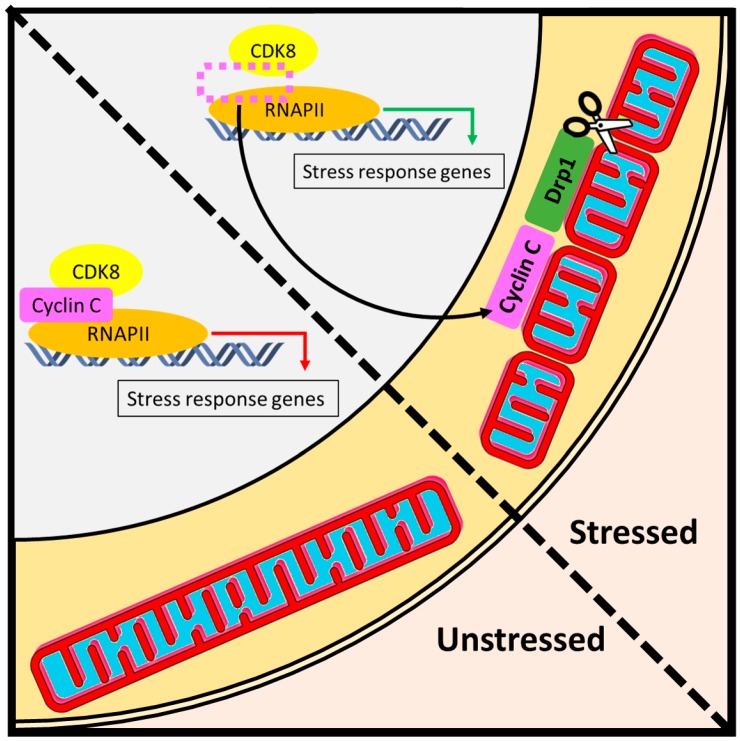
Day job versus night job function of cyclin C. (**Left**) Under normal conditions, cyclin C performs its day job as a nuclear transcription factor that regulates RNA Pol II-dependent activity. In unstressed cells, mitochondria maintain a reticular structure [33]. (**Right**) The night job function of cyclin C is initiated by cellular stress and is associated with the activation of two layers of stress defense. First, under adverse conditions such as oxidative stress, a portion of cyclin C exits the nucleus to induce Drp1-dependent mitochondrial scission. The resulting fragmentation of mitochondrial network facilitates downstream stress responses such as mitophagy or apoptosis. Second, upon dissociation of cyclin C from the promoters of stress response genes, the activity of RNA Pol II is regained, which triggers a host of adaptive mechanisms such as antioxidant defense or unfolded protein response.

**Figure 6 biology-08-00003-f006:**
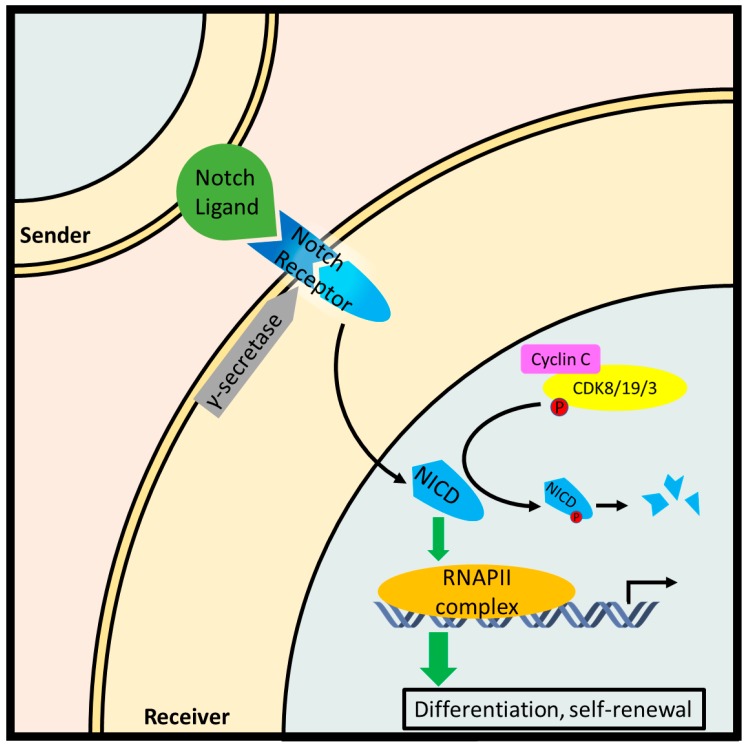
Cyclin C-Cdk8/19/3 inhibit the Notch pathway by phosphorylating the Notch intracellular fragment (NICD) and promoting its degradation in leukemic T cells. In canonical Notch signaling, a contiguous cell exposes a Notch ligand which is recognized by a Notch receptor paralog on a receiving cell. Subsequently, a two-step proteolytic cleavage produces NICD which stimulates with a transcriptionally paused RNA Pol II complex resulting in the downstream expression of genes involved in stem cell differentiation and self-renewal.

**Table 1 biology-08-00003-t001:** The human cyclin family. Cyclins preferentially bind a specific Cdk, e.g., cyclin C is the regulatory subunit for Cdk8 although some promiscuity has been observed. Cyclin-Cdk kinases comprise cell cycle regulators (Class I), transcription factors (Class II), and others with unspecific functions (Class III) [6,7].

Class	Function	Cyclin	Gene	Cyclin-Dependent Kinase
I	cell cycle	A, B	*CCNA*, *CCNB*	1–3
	regulation	D	*CCND*	4–6, 14–18
		E	*CCNE*	1–3
		F, J, O	*CCNF*, *CCNJ*, *CCNO*	2
		G, I	*CCNG*, *CCNI*	5
II	transcriptional	C	*CCNC*	8, 19
	regulation	H	*CCNH*	7, 20
		K	*CCNK*	9, 12, 13
		L, M	*CCNL*, *CCNM*	10, 11
		T	*CCNT*	9
III	other	Y	*CCNY*	5, 14–18

**Table 2 biology-08-00003-t002:** Examples of Cdk8 inhibitors with reported anti-cancer properties.

Compound	Cancer	Reference
BIRB796	NS ^1^	[48,111]
CCT251545	AML, colorectal	[112,113,114]
CCT251921	colorectal	[114,115]
Compounds 1–5, 7–11	NS ^1^	[111]
Compound 21	colon	[116]
Compound 32	colon	[117]
Compound 42	colorectal	[114,115]
Cortistatin A	AML ^2^	[118,119]
Imatinib	NS ^1^	[48]
MSC253081	colon, colorectal	[114,120]
SEL120-34A	AML ^2^	[113]
Senexin A	breast, CLL ^3^, colon, colorectal, fibrosarcoma, lung	[121,122,123,124]
Senexin B	AML, breast, colon, colorectal	[113,124,125]
Sorafenib	NS ^1^	[48,111]

^1^ NS, not specified; Cdk8 inhibition was not directly studied in a cancer model. ^2^ AML, acute myeloid leukemia. ^3^ CLL, chronic lymphocytic leukemia.

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
