# Peer review of "Cyclin C: The Story of a Non-Cycling Cyclin"

_biology, 2019, doi:10.3390/biology8010003_

Reviewer 1 Report

Jezek et al. review the transcriptional and non-transcriptional roles of Cyclin C in tumor suppression. Cyclin C is known to interact with Mediator and to leave the nucleus upon oxidative stress to induce mitochondrial fission/fragmentation by directly activating the GTPase Drp1/Dnm1. Cyclin C as a potential pharmacologic target is also discussed.

Except in a few places, the language is generally fine but the major sections and many sentences seem somewhat disconnected. Thus reorganizing or including more transitions between the major sections and subsections would be helpful to generate a more logical flow of ideas. In addition, the authors should thoroughly re-edit to improve impact of an otherwise very nice and comprehensive review. By removing some redundancies and tightening/clarifying the language, the document could be shortened or create more space for insufficiently described sections.

Specific comments

1.     The topics discussed in the major subdivisions of this review seem out of order, or could be better integrated.

2.     Abstract: “activates the mitochondrial fission GTPase Drp1”, is missing a word or needs rewording.

3.     It would be helpful to clarify at the start of the Abstract and Introduction that there are two/three major classes of cyclins, the better known regulators of the cell cycle and non-cycling transcription factors, without waiting until half-way through the Introduction.

4.     Sentences on lines 52-55 could be better connected to the previous sentence – one example.

5.     Is cyclin C unique among class II cyclins by being non-cycling, or the first class II to be demonstrated (Lines 55-59) – please clarify. If the authors could thoroughly review the organization of the sentences and the connections between sentences throughout, re-editing would be beneficial. Only a few examples noted here.

6.     The official gene name for cyclin C is not mentioned until line 83, but perhaps could be included for all the cyclins in Table 1 (which could be labeled “Table 1”, or is this found on the previous page?).

7.     In section 2.1, a diagram illustrating that both classes consist of little more a Cycin_N and a Cyclin_C domain fold, and their relative sequence similarity would be helpful.

8.     The tree in Fig 3 is difficult to read and lacks a scale or obvious root – please explain and/or use a 2-column width.

9.     Line 88, clarify “In this two-pronged scenario”, as there are several potential dichotomies, and at the end of this paragraph is another: “transcription-dependent and transcription-independent”.

10.  Missing word line 90, “direction action of ___ outside of nucleus”.

11.  It appears that many different ideas are compressed into the last half of the paragraph on lines 90-95. This could be clarified and possibly separated into a separate paragraph.

12.  Highlighted amino acid sequence features in Fig 1 are not mentioned in the text in section 2.1, and the amino acids of cyclin C discussed in section 5.1 are not shown in Fig 1.

13.  The interesting phylogenetic analysis section 2.3 leaves one hanging. If not resolvable based on current knowledge, the authors should acknowledge or explain current attempts at such. Explain or offer potential explanations for their common ancestry. It was a great surprise many years ago that the cell still cycles after deletion of the major cell cycle cyclins, leading to the notion that cyclins have critical roles in integrating cell cycle events over driving cell cycle events.

14.  Typo line 199 “play”.

15.  Problem on lines 226, and 227; the sentence on 227-228, and the next sentence appears out of order. This paragraph is a bit rough, but section 4 generally contains redundancies and is not well organized. For example, while there is ample material to connect human/yeast Drp1/Dnm1 discussed in section 4.1 with cell death, these factors are not discussed in the very brief section 4.2 on cell death. This brevity of section 4.2 leaves the reader asking many questions [e.g. does cyclin C movement to mitochondria cause caspase 3-mediated DNA cleavage (apoptosis) in mammals and yeast].

16.  There is an apparent inconsistency in the number of subunits in the yeast Mediator complex, 21 (line 320) or 25 (line 322).

17.  Can the authors make clear what is known about the specific stress or stress signaling that causes Cyclin C movement to mitochondria, and what activates cyclin C/releases it from Med13 anchor?

18.  Table 2 is a nice addition to section 8, but is not discussed in the text.

19.  Missing reference line 503-506.

Author Response

1.     The topics discussed in the major subdivisions of this review seem out of order, or could be better integrated.

To better facilitate the flow of ideas, we changed the order of sections 2.2 and 2.3.

2.     Abstract: “activates the mitochondrial fission GTPase Drp1”, is missing a word or needs rewording.

The phrase was changed to: “activates the guanosine 5’-triphosphatase (GTPase) Drp1”.

3.     It would be helpful to clarify at the start of the Abstract and Introduction that there are two/three major classes of cyclins, the better known regulators of the cell cycle and non-cycling transcription factors, without waiting until half-way through the Introduction.

Cell cycle regulating and non-transcriptional cyclins were highlighted as class I and II, respectively, in the Abstract and Introduction.

4.     Sentences on lines 52-55 could be better connected to the previous sentence – one example.

This paragraph was simplified to improve its connection with the preceding text.

5.     Is cyclin C unique among class II cyclins by being non-cycling, or the first class II to be demonstrated (Lines 55-59) – please clarify. If the authors could thoroughly review the organization of the sentences and the connections between sentences throughout, re-editing would be beneficial. Only a few examples noted here.

We underscored in the text that cyclin C is not a unique non-cycling cyclin but was the first discovered transcriptional cyclin that showed a reduced cycling pattern. We rephrased other sentences as well.

6.     The official gene name for cyclin C is not mentioned until line 83, but perhaps could be included for all the cyclins in Table 1 (which could be labeled “Table 1”, or is this found on the previous page?).

Gene names were included in Table 1 for all the cyclins. Table 1 title and description is now on common page with the table body.

7.     In section 2.1, a diagram illustrating that both classes consist of little more a Cycin_N and a Cyclin_C domain fold, and their relative sequence similarity would be helpful.

We introduced a diagram illustrating domain composition of class I and II cyclins and the sequence identity and similarity between cyclin C and other cyclins as part of a new figure (Fig. 1).

8.     The tree in Fig 3 is difficult to read and lacks a scale or obvious root – please explain and/or use a 2-column width.

Fig. 2 (former Fig. 3) as well as all other figures were resized. A scale bar was added, and the root node was enlarged and marked by a short line segment.

9.     Line 88, clarify “In this two-pronged scenario”, as there are several potential dichotomies, and at the end of this paragraph is another: “transcription-dependent and transcription-independent”.

The word “two-pronged” was omitted from the sentence and “transcription-dependent and transcription-independent” was replaced by “dual role” to make the meaning unequivocal.

10.  Missing word line 90, “direction action of ___ outside of nucleus”.

This sentence was rephrased.

11.  It appears that many different ideas are compressed into the last half of the paragraph on lines 90-95. This could be clarified and possibly separated into a separate paragraph.

This section rephrased and separated into a new paragraph.

12.  Highlighted amino acid sequence features in Fig 1 are not mentioned in the text in section 2.1, and the amino acids of cyclin C discussed in section 5.1 are not shown in Fig 1.

The reference to Fig. 4 (former Fig. 1) was deleted from section 2.1 as this chapter is intended to describe primary sequence features of the cyclin family as a whole. Characteristics of the cyclin C sequence are developed in section 2.3 (former section 2.2), containing a reference to Fig. 4. A brief mention of the aC a-helix of Cdk8 was inserted so that all sequence features from Fig. 4 are now mentioned in this section.  Residues from the putative Med12- and Med13-binding groove of cyclin C (I33, R49, W160, D165, Y167) are now shown in both Figs. 3 and 4.

13.  The interesting phylogenetic analysis section 2.3 leaves one hanging. If not resolvable based on current knowledge, the authors should acknowledge or explain current attempts at such. Explain or offer potential explanations for their common ancestry. It was a great surprise many years ago that the cell still cycles after deletion of the major cell cycle cyclins, leading to the notion that cyclins have critical roles in integrating cell cycle events over driving cell cycle events.

The common ancestry of the transcriptional cyclins K and T and the cell cycle cyclins A, D, and E may be potentially explained by redundancy that these regulators may have in engaging the cell cycle-specific Cdks [Sherr CJ, Roberts JM 2004 Genes Dev 18:2699]. One may speculate that the ability to “cycle” was originally acquired from ancient predecessor sequences of cyclin K and T but this function was superseded by cyclins A, D, and E during evolution. We included this possibility as one of the potential explanations in the text.

14.  Typo line 199 “play”.

This was corrected.

15.  Problem on lines 226, and 227; the sentence on 227-228, and the next sentence appears out of order. This paragraph is a bit rough, but section 4 generally contains redundancies and is not well organized. For example, while there is ample material to connect human/yeast Drp1/Dnm1 discussed in section 4.1 with cell death, these factors are not discussed in the very brief section 4.2 on cell death. This brevity of section 4.2 leaves the reader asking many questions [e.g. does cyclin C movement to mitochondria cause caspase 3-mediated DNA cleavage (apoptosis) in mammals and yeast].

This part of sentence was deleted, the order of the following sequences was changed, and the paragraph was rephrased to improve the logical flow of the paragraph. We restructured these two sections so that only section 4.2 is fully focused on the role of cyclin C in modulating apoptotic cell death. Given that apoptosis is a complex process, there are still many unresolved question within this field. Experiments are currently underway in the mammalian system to determine at which particular steps cyclin C comes into play to fill in the gap in our knowledge of cell death paradigms. A brief statement informing the reader that the mechanism of cyclin C-induced apoptosis deserves further investigation was appended at the end of this section.

16.  There is an apparent inconsistency in the number of subunits in the yeast Mediator complex, 21 (line 320) or 25 (line 322).

We replaced the number “21” with the word “multi-subunit” to correct for the inconsistency.

17.  Can the authors make clear what is known about the specific stress or stress signaling that causes Cyclin C movement to mitochondria, and what activates cyclin C/releases it from Med13 anchor?

In yeast, the cyclin C-dependent signaling is initiated by the cell-wall sensors Mtl1, Wsc1, and Mid2. Upon activation by specific cues such as nutrient deprivation, oxidative, ethanol, heat shock or osmotic stress, these receptors initiate signal transduction through the cell wall integrity (CWI) MAP kinase pathway to regulate the stability of Med13, which is part of the Cdk8-dependent kinase module and serves as the nuclear anchor for cyclin C. Phosphorylation of Med13 by the MAP kinase Slt2 induces SCF ubiquitin ligase-dependent proteasomal destruction of Med13 and this, in turn, facilitates the translocation of cyclin C into the cytosol to induce mitochondrial fission. In higher eukaryotes, the stress sensing mechanisms that lead to the activation of cyclin C have not been elucidated yet. These important paradigms of cyclin C biology are described as part of section 4.1.

18.  Table 2 is a nice addition to section 8, but is not discussed in the text.

We added a brief discussion on the mainstream pharmacological inhibitors of Cdk8 in the respective section.

19.  Missing reference line 503-506.

Proper reference was added.

Reviewer 2 Report

This is a comprehensive and nice review on the functions of Cyclin C. This cyclin is much less known compared to the others but still is quite interesting. The authors do a good job in providing a balanced view of all the information about Cyclin C.

This is a good manuscript but the authors should address the following issues:

The section 4.2. on apoptosis seems out of place and does not seem very important. If this would be my manuscript, I probably would omit this section.

The paper on Cyclin C/Cdk3->pRb [57] is mentioned twice. The results from this paper have never been reproduced by anybody else despite that it has been published 14 years ago.      Therefore, either these results are not that important or potentially not reproducible. I would suggest that the authors use this reference with caution.

Author Response

The section 4.2. on apoptosis seems out of place and does not seem very important. If this would be my manuscript, I probably would omit this section.

We kept and improved the section on apoptosis as there is a new data emerging that cyclin C directly activates the apoptotic effector Bax in mammalian cell lines. A research manuscript on cyclin C-mediated Bax activation was submitted for publication during the preparation of this review and is currently under consideration in EMBO Reports, nevertheless acceptance of this article is past the deadline for resubmission of the present review.

The paper on Cyclin C/Cdk3->pRb [57] is mentioned twice. The results from this paper have never been reproduced by anybody else despite that it has been published 14 years ago.      Therefore, either these results are not that important or potentially not reproducible. I would suggest that the authors use this reference with caution.

We deleted this reference as well as reference 58 (cyclin C regulates human hematopoietic stem/progenitor cell quiescence) and the relevant paragraphs.

Reviewer 3 Report

This is a comprehensive, well written manuscript. The authors adequately covered the topic. This reviewer  has no reservations to any part of this review.  

Author Response

No questions or comments.